# Surface Complexation Modeling of Biomolecule Adsorptions onto Titania

**Nataliya N. Vlasova * and Olga V. Markitan**

Chuiko Institute of Surface Chemistry, National Academy of Sciences of Ukraine, General Naumov Str., 17, Kyiv 03164, Ukraine; kammar@ukr.net
* Correspondence: natalie.vlasova@gmail.com; Tel.: +38-050-385-21-75

**Abstract:** The adsorption of nucleic acid components on the surface of nanocrystalline titanium dioxide (anatase, $pH_{pzc}$ = 6.5) in NaCl solutions was investigated using potentiometric titrations and multibatch adsorption experiments over a wide range of pH and ionic strengths. The Basic Stern surface complexation model was applied to experimental data to obtain quantitative equilibrium reaction constants. Adsorption results suggest that there is a considerable difference in the binding of nucleobases, nucleosides, and nucleotides with an anatase surface.

**Keywords:** surface complexation model; titanium dioxide; nucleic acid components

## 1. Introduction

The interface between biomolecules and inorganic oxide surfaces has attracted considerable attention as a decisive factor in bio-applications of nanostructured oxides [1–6]. An important inorganic oxide with unique properties is titanium dioxide. Titania is a versatile material with specific optical, environmental, and photocatalytic properties [7,8] used in various applications such as biomedicine and cosmetics [9]. Understanding the interaction at the nano–bio interface plays a key role in the development of nanoscience and nanotechnology.

Defining the interaction of oxide particles with a biological medium is an extremely complex task. Such interactions are very complicated because they include the formation of many different-in-nature bonds involving numerous groups of biomolecules and surface functional groups of the solid. Studies of the interaction of nano-oxide surfaces with monomeric biomolecules such as nucleotides and amino acids, from which the corresponding biopolymers, nucleic acids, and proteins are formed, can serve as a basis for establishing the mechanism at the molecular level. The investigation of amino acid adsorption at titanium oxide surfaces has received significant interest due to its relevance in several fields of chemistry, biology, and medicine [10–16]. Investigations have usually focused on the identification of adsorbing species, their structure, and the types of interactions that take place between the adsorbing molecules and the surface. As for nucleic acid components, most of publications are devoted to the study of their interaction with clay minerals [17–19] and oxides [20–22] other than titania [23].

The aim of this paper is to study of adsorption of nucleic acid components at the titanium oxide/aqueous solution interface in terms of surface complexation theory. This approach allows for the quantitative determination of the stability of surface complexes and the prediction of the types of interactions between adsorbing molecules and active sites of a solid.

## 2. Materials and Methods

A nanocrystalline titanium dioxide (nanopowder, Aldrich) with a specific surface area $62 \pm 5 \, m^2/g$ (Nova 1200, Quantachrome) was used. According to X-ray diffraction data (Dron-3M diffractometer),

the titanium dioxide was anatase (ICPDS, 84-1286). The crystallite size calculated by the Sherrer equation was 30 nm, which corresponded to the manufacturer's data (<25 nm).

## 2.1. Potentiometric Titration

Potentiometric titrations of the titania suspension (10 g/L) with acid and base were performed at $25 \pm 0.05$ °C in gas-tight, 50-mL centrifuge polyethylene tubes in a thermostated water bath with a shaker bubbled with Ar. The ionic strength was adjusted by a background electrolyte to 0.01, 0.05, and 0.1 M using NaCl (Merck, p.a.). The pH of the suspension was measured using an Inolab Level 2P pH meter (WTW) equipped with a combined electrode (SenTix81) and temperature probe. The electrode was calibrated using a three-point calibration with a commercial pH buffer (CertiPure, Merck) to a precision of $\pm 0.02$ units. The titrations were carried out in batch mode by discrete additions of dilute HCl or NaOH prepared from standard solutions (Titrisol, Merck) in deionized water under bubbling with Ar. Electrode readings were taken when a drift less than 0.002 pH units in 10 min was attained, with a minimum reading time of 10 min between additions.

## 2.2. Sorption Experiments

For adsorption experiments, nucleic acid components were used as received: bases—adenine, guanine, cytosine, and uracil (Fluka, p.a.); nucleosides—adenosine, $2'$-deoxyadenosine, guanosine, cytidine, 2-deoxycytidine, and uridine (Fluka, p.a.); pyrimidine nucleotides as sodium salts of cytidine-$5'$-monophosphate (CMP) (Reanal, p.a.), uridine-$5'$-monophosphate (UMP) (Reanal, p.a.), and orotidine-$5'$-monophosphate (OMP) (Sigma-Aldrich, p.a.); and purine nucleotides as sodium salts of adenosine-$5'$-monophosphate (AMP) (Reanal, p.a.), guanosine-$5'$-monophosphate (GMP) (Reanal, p.a.), and inosine-$5'$-monophosphate (IMP) (Sigma, p.a.).

Nucleic acid component adsorption was studied at $22 \pm 1$ °C. Equal volumes of titania suspension (2 or 10 g/L) and component solutions (0.2 mmol/L) were mixed in centrifuge tubes. A constant ionic strength was maintained by addition of background electrolyte NaCl (0.001, 0.01, and 0.1 M). The pH was adjusted to the desired value between 3 and 9 with HCl or NaOH solutions. All suspensions were stirred for 2 h, the final pH was measured, and then the solid phase was separated by centrifugation (8000 rpm, 20 min). It was preliminarily found that 2 h was sufficient time to reach adsorption equilibrium. Nucleic acid component concentrations were determined from the absorption spectra in the UV region (Specord M-40 spectrophotometer, Carl Zeiss Jena). All components were characterized by the absorption bands at 260–270 nm. The pH dependences of the positions and intensities of the bands were determined beforehand.

The amounts of adsorbed nucleic acid components (in mol/g or as % of adsorption) were calculated as the difference between initial and equilibrium concentrations. Experimental adsorption values are shown in the figures as symbols, and the calculated adsorption curves are shown as lines.

## 2.3. Model Calculations

The Basic Stern surface complexation model [24] and the GRFIT software [25] were used for quantitative interpretation of experimental data. The GRFIT program is very convenient because the fitting of adjustable parameters is accompanied by a graphic drawing of an adsorption curve, from which one can immediately judge how successful the choice of reaction equations and initial values of the adjustable parameters is. A first step of the work was the choice of components of the solution, solid, and charge on different planes of the interface. The components of the solution were initial species of biomolecule, proton, and electrolyte ions. The surface components were considered the initial hydroxyl group (TiOH) and the electrical components (exp0 and exp1), which correspond to the values of the charges of the species in the 0 and 1 planes of the interface. They were expressed as coefficients of the potentials at the respective planes of the electric double layer (EDL). The next step was to create a matrix in which the species present in the solution (all forms of biomolecules other than the initial one) and surface species (protonated and ionized titania hydroxyl group), ion pairs with

background electrolyte ions, and surface complexes with biomolecules were defined as combinations of components.

It should also be noted that the program requires knowledge of the characteristics of a solid such as the specific surface area and concentration of functional groups or site density. The concentration of functional groups can be set as an adjustable parameter or is selected on the basis of data (for example, acid–base titration of the oxide suspension).

## 3. Results and Discussion

### 3.1. Titania Surface Acidity

The charging of an oxide surface as a result of its interaction with protons of an aqueous solution gives rise to an electrical double layer between the electrolyte solution and the oxide surface, which in turn affects complexation reactions with inorganic and organic molecules. Surface complexation models have been successfully used to describe the binding of protons and chemical substances present in a solution by active groups of oxide surfaces [24,26–28]. Regardless, the study of the adsorption interaction of organic or inorganic compounds dissolved in water must be preceded by a detailed investigation of the protolytic properties of oxide surface groups.

The acid–base properties of titania and adsorption equilibria with organic molecules were quantitatively estimated using the Basic Stern model of surface complexation. According to this model, the interface is divided into two regions: the compact region (extending up to several angstroms from surface) and the diffuse one, where the counterions accumulate to compensate the surface charge. The compact region involves two charged planes: the surface (or zero plane) of the hydroxo-surface groups and the head end of the diffuse layer of the interface (or plane 1). Potential-determining ions are adsorbed in one of them (zero plane), while weakly bound counterions are adsorbed in another plane (plane 1). These counterions form ions pairs with the surface hydroxyl groups of opposite charge.

The results of potentiometric titration of titania suspensions were used to calculate surface charge $\sigma_0$ (C/m$^2$) by the following equation:

$$\sigma_0 = (F/Sm)((V_0 + V_A + V_B)([OH^- - [H^+]) + V_A A - V_B B) \tag{1}$$

where $F$ is the Faraday constant (96,485 C/mol), $S$ is the specific surface area (m$^2$/g), $m$ is the amount of titania (g) in initial volume $V_0$ of suspension taken for titration (L), and $V_A$ and $V_B$ are the volumes (l) of added acid and base with concentrations $A$ and $B$ (mol/L), respectively.

Figure 1 shows the surface charge of titanium dioxide as a function of pH and ionic strength. These data demonstrate that all curves had an intersection point at pH 6.5, which was the point of zero charge. This value is in good agreement with data published for anatase [29].

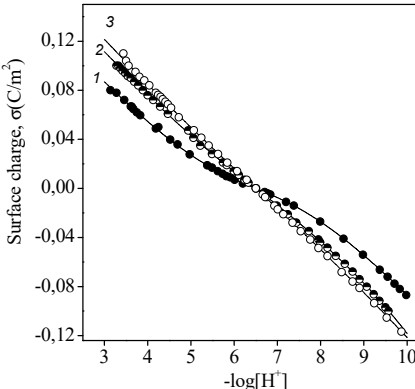

**Figure 1.** Surface charge as a function of pH and ionic strength: experimentally measured values (symbols) and those calculated in terms of the Basic Stern model (lines) for titanium dioxide. C$_{TiO2}$ = 10 g/L, C$_{NaCl}$ = 0.001 (*1*), 0.05 (*2*), and 0.1 (*3*) M.

The interactions between oxide active sites and aqueous protons cause either protonation or ionization reactions:

$$\equiv \text{TiOH} + \text{H}^+ \leftrightarrow \equiv \text{TiOH}_2^+ \tag{2}$$

$$\equiv\text{TiOH} \leftrightarrow \equiv\text{TiO}^- + \text{H}^+. \tag{3}$$

These equilibria are characterized by the corresponding constants:

$$K_{S1}^{\text{int}} = \frac{[TiOH_2^+]}{[TiOH][H_S^+]} \text{ and } K_{S2}^{\text{int}} = \frac{[TiO^-][H_S^+]}{[TiOH]}$$

where [TiOH], [TiOH$_2^+$], and [TiO$^-$] are the equilibrium concentrations (mol/l) of neutral, protonated, and ionized hydroxyl groups on the oxide surface. Surface concentrations of protons [H$_S^+$] in plane 0 are related to the proton concentration in the bulk solution via the Boltzmann distribution law:

$$[H_S^+] = [H^+] \exp(-F\Psi_0/RT)$$

where $\Psi_0$ is the potential value in plane 0 of the electric double layer. Hence,

$$K_{S1}^{\text{int}} = \frac{[TiOH_2^+]}{[TiOH][H^+]} \exp(F\Psi_0/RT)$$

$$K_{S2}^{\text{int}} = \frac{[TiO^-][H^+]}{[TiOH]} \exp(-F\Psi_0/RT)$$

It has been established that background electrolyte ions form complexes with oxide hydroxyl groups [26,27]. It is very likely that at least one layer of water molecules is located between weakly sorbed ions and surface sites. This layer separates the ions from oxygen or metal ions of the surface (i.e., they form ion pairs or outer-sphere complexes). Therefore, when describing the protolytic properties of an oxide, the formation of outer-sphere complexes with electrolyte ions must be taken into account in addition to protonation and ionization reactions:

$$\equiv \text{TiOH} + \text{H}^+ + \text{Cl}^- \leftrightarrow \equiv \text{TiOH}_2^+\text{Cl}^- \tag{4}$$

$$K_{Cl^-}^{\text{int}} = \frac{[TiOH_2^+Cl^-]}{[TiOH][H_S^+][Cl^-]} = \frac{[TiOH_2^+Cl]}{[TiOH][H^+][Cl^-]} \exp\left(\frac{F(\Psi_0 - \Psi_1)}{RT}\right);$$

$$\equiv\text{TiOH} + \text{Na}^+ \leftrightarrow \equiv\text{TiO}^-\text{Na}^+ + \text{H}^+ \tag{5}$$

$$K_{M^+}^{\text{int}} = \frac{[TiO^- - Na^+][H_S^+]_S}{[TiOH][Na_S^+]} = \frac{[TiO^- - Na^+][H^+]}{[TiOH][Na^+]} \exp\left(\frac{F(\Psi_1 - \Psi_0)}{RT}\right).$$

From the reactions taking place on the titania surface in the presence of an aqueous electrolyte solution, the surface charge density may be expressed by the following equation:

$$\sigma_0 = F/mS([TiOH_2^+] + [TiOH_2^+Cl^-] - [TiO^-] - [TiO^-Na^+].$$

Thus, the equilibrium reaction constants of protonation, ionization, and ion pair formation reactions were determined by comparing the surface charge values calculated from the potentiometric titration by Equation (1) and the values fitted using GRFIT software as a result of successive approximations. The best agreement between experimental data and calculated curves was achieved for the set of parameters listed below.

1.  The concentration of hydroxyl groups on the surface was 0.5 mmol/g, which corresponded to the density of active site per surface unit of 5 groups/nm$^2$. The value for different crystallographic

faces of titanium dioxide was found to be 5.2–7.0 groups/nm$^2$ [29], although somewhat smaller values are usually determined experimentally [30–33].

2. The specific capacitance of EDL was estimated to be 0.76 F/m$^2$. As has been shown in [34], such a low capacitance is quite feasible and has a physical sense. This fact indicates that the first layer of physically adsorbed water has a relatively low dielectric permittivity, which is virtually independent of the dielectric properties of the solid [35].

3. The equilibrium reaction constants (with an accuracy ±0.05) for the aforementioned reactions were as follows: protonation (2), $\log K_{S1}^{int} = 5.2$; ionization (3), $\log K_{S2}^{int} = -7.8$; anion binding (4), $\log K_{Cl}^{int} = 6.2$; and cation binding (5), $\log K_{Na}^{int} = -6.8$.

Figure 1 shows both the experimental points and lines calculated for the pH dependences of the surface charge.

### 3.2. Adsorption of Nucleic Acid Components

#### 3.2.1. Nucleobases

In an aqueous solution, purine and pyrimidine bases undergo protolytic and tautomeric transformation. The base formulas are shown in Table 1 as ketone (lactam) tautomers because they prevail in aqueous solutions [36,37]. All examined bases besides uracil existed in the acid solution as cations. While the pH increased to a value higher than $pK_1$, these compounds existed in the solution as neutral molecules, which were transformed into anions when the pH was above $pK_2$. Uracil was not protonated in the acid environment; in the alkaline medium (at pH $\geq$ 10), it was ionized into an anionic form.

**Table 1.** Ionization constants of nucleic acid bases.

| Base | Ionization Reaction (p$K$) [38] (0.01 M) | |
|---|---|---|
| | $BH^+ \leftrightarrow B + H^+$ | $B \leftrightarrow B^- + H^+$ |
|  adenine | 3.92 (N1H$^+$) | 9.63 (N9H) |
|  guanine | 2. 20 (N1H$^+$) | 9.38 (N9H) |
|  cytosine | 4.23 (N3H$^+$) | 12.20 (N1H) |
|  uracil | | 10.13 (N1H) |

Figure 2 shows the adsorption of the bases from the aqueous 0.01 M NaCl solution onto titanium dioxide surface in dependence on pH. According to the observation, the bases can be arranged as follows: guanine > cytosine > adenine > uracil; that is, the molecules that contained amino and ketone groups were adsorbed to a greater extent than those containing only an amino or a ketone group. Adsorptions of guanine, cytosine, and adenine significantly increased with pH. It may be assumed that neutral molecules were adsorbed more readily than cations. This is quite reasonable because at pH < 6.5, the titania surface was as a whole positively charged; therefore, electrostatic repulsion was possible between the similarly charged nucleobase cations and the functional surface groups.

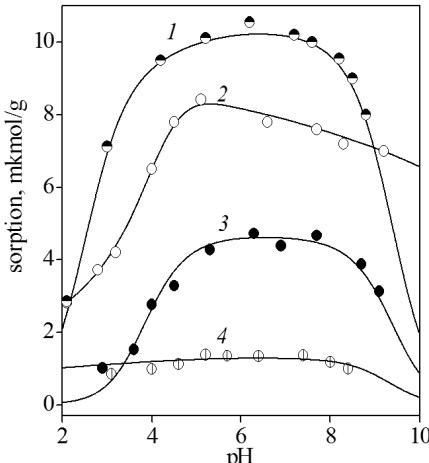

**Figure 2.** Adsorption of guanine (*1*), cytosine (*2*), adenine (*3*), and uracil (*4*) from 0.01 M NaCl solution onto titatia surface: $C_{TiO2}$ = 5 g/L, and $C_{base}$ = 0.1 mmol/L.

The considerable decreasing of the adsorption values of guanine and adenine at pH > 8 can be explained by the formation of anions in the solution, which were electrostatically repelled from negatively charged titania surface groups. For the same reason, adsorption of uracil, which was almost constant up to a pH of ~8, somewhat decreased as the pH further increased.

It is noteworthy that the adsorption of all studied bases was independent of ionic strength. As the background electrolyte concentration increased to 0.1 M, the adsorption of cytosine and adenine remained virtually equal to those in the presence of 0.01 M NaCl. This fact indicates the absence of competition between electrolyte cations and base molecules. Since electrolyte cations form electrostatic ion pairs with the functional groups, one may assume that base molecules interact with surface groups via hydrogen bonding or dispersion interaction. Taking into account the composition of the titania surface layer, which, depending on pH, contains protonated, neutral, or ionized groups, the occurrence of the following surface complexation reactions are suggested:

$$\equiv TiOH_2^+ + B \leftrightarrow \equiv TiOH_2^+ \cdots B \tag{6}$$

$$\equiv TiOH + B \leftrightarrow \equiv TiOH \cdots B \tag{7}$$

$$\equiv TiO^- + B \leftrightarrow \equiv TiO^- \cdots B \tag{8}$$

$$\equiv TiOH + BH^+ \leftrightarrow \equiv TiOH \cdots BH^+ \tag{9}$$

$$\equiv TiO^- + BH^+ \leftrightarrow \equiv TiO^- \, BH^+ \tag{10}$$

where B and $BH^+$ are the neutral and cationic forms of nucleobases, respectively.

The probability of reaction (10) was very low because ionized groups emerged on the surface at pH > 6.5, while the solution contained almost no cations.

The equilibrium constants of the complexation reactions were calculated using the GRFIT program. It turned out that guanine, adenine, and uracil adsorption was quantitatively described by reaction (7); that is, the interaction between neutral base molecules and neutral groups of titania surface occurs. For cytosine, the complexation reaction (9) between neutral TiOH groups and base cations had to be taken into account. All calculated formation constants of the surface complexes are listed in Table 2.

**Table 2.** Formation constants of nucleobase and nucleoside surface complexes (logK ± 0.05).

| Biomolecule | Surface Complex | |
|:---:|:---:|:---:|
| | $\equiv$TiOH$\cdots$BH$^+$ | $\equiv$TiOH$\cdots$B |
| adenine | | 2.38 |
| adenosine | | 2.35 |
| 2′-deoxyadenosine | | 1.75 |
| guanine | | 2.76 |
| guanosine | | 2.71 |
| cytosine | 3.60 | 2.50 |
| cytidine | 3.50 | 2.43 |
| 2′-deoxycytidine | 3.38 | 2.13 |
| uracil | | 1.71 |

### 3.2.2. Nucleosides

Nucleosides are N-glycosides of the bases, in which one carbon atom (C1) of pentose is bonded to the N1 atom of the pyrimidine cycle or the N9 atom of the purine cycle via a glycosidic bond. There are two series of nucleosides: ribonucleosides, which contain D-ribose as the sugar component, and 2′-deoxyribonucleotides containing deoxy-D-ribose.

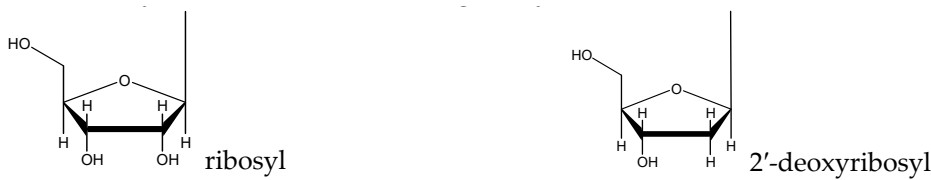

Figure 3 shows the adsorption of ribo- and deoxyribonucleosides of the corresponding bases. The changes in adsorption of ribonucleosides approximately corresponded to the same row as that for the bases: guanosine > cytidine > adenosine. Adsorption of uridine is not presented in the figure because its values were within the concentration measurement error in all pH ranges. The quantitative parameters of nucleoside adsorption were obtained under the assumption that the same complex formation reactions (7) and (9) are carried out as those for bases. The formation constants calculated for the surface complexes are listed in Table 2.

The data shown in Figure 3 indicate that nucleoside adsorption was approximately equal to that of the bases, although nucleoside molecules contain additional sites capable of binding with surface groups (hydroxyl groups of ribosyl residue). The heterocyclic base molecules (pyrimidine and purine) are known to have an almost planar structure, while the residues of saturated cyclic compounds have a bent conformation [36]. Obviously, an arrangement of nucleoside molecules in which all functional groups of heterocycles and ribosyl moiety could be in contact with the surface groups to form hydrogen bonds can hardly be achieved. However, the group at the 2′-position of ribosyl may play a certain role in binding with the surface, which was evident from the adsorption values of deoxyribonucleosides being lower than those of ribonucleosides.

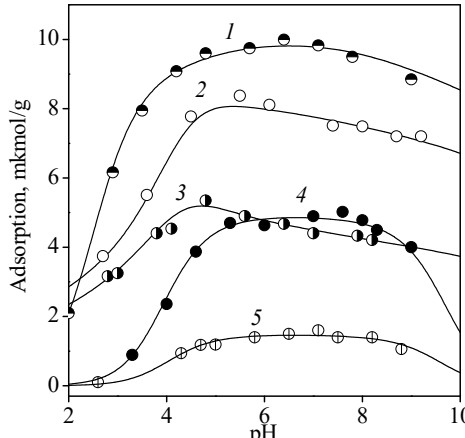

**Figure 3.** Adsorption of guanosine (*1*), cytidine (*2*), adenosine (*3*), 2′-deoxycytidine (*4*), and 2′-deoxyadenine (*5*) from 0.01 M NaCl solution onto titanium dioxide surface: $C_{TiO2}$ = 5 g/L, and $C_{nucleosine}$ = 0.1 mmol/L.

### 3.2.3. Nucleotides

Nucleotides, which are repeating monomer units in nucleic acids, are phosphorylated nucleosides. This means that one of the carbohydrate hydroxyl group is bonded with phosphoric acid residue. Phosphate groups in nucleoside monophosphates are characterized by two ionization constants: a primary group dissociates at pH < 1–2, while the secondary groups are ionized at pH > 6 [36]. The formulas and ionization constants of all nucleotides under investigation are shown in Table 3.

**Table 3.** Ionization constants of nucleotides [39].

| Nucleotide | Ionization Constant $pK_n$ (0.01 M) |
|:---:|:---:|
| Cytidine-5′-monophosphate ($H_2L^{\pm}$), CMP | 4.31 ($N_3$–$H^+$) 6.15 (–$PO_3H^-$) |
| Uridine-5′-monophosphate ($HL^-$), UMP | 6.04 (–$PO_3H^-$) 9.39 ($N_3$–H) |
| Orotidine-5′-monophosphate ($H_2L^-$), OMP | 2.40 (–COOH) 6.10 (–$PO_3H^-$) |

**Table 3.** *Cont.*

| Nucleotide | Ionization Constant p$K_n$ (0.01 M) |
|---|---|
| Inosine-5′-monophosphate (HL⁻), IMP | 6.47 (–PO₃H⁻) |
| Guanosine-5′-monophosphate (H₂L±), GMP | 2.48 (–N₇H⁺) 6.48 (–PO₃H⁻) |
| Adenosine-5′-monophosphate (H₂L±), AMP | 3.96 (–N₁H⁺) 6.46 (–PO₃H⁻) |

In the CMP molecule, a nitrogen atom N3 of the pyrimidine ring was protonated in the acid solution, with proton release occurring at pH > 4. Depending on pH, CMP existed in the solution as its completely protonated form $H_3L^+$, a zwitterion $H_2L^{\pm}$ (after first phosphate proton dissociation), monoanion $HL^-$ (ionization of protonated group N3-H⁺), and dianion $L^{2-}$ (second phosphate proton release). Since phosphate groups are ionized at low pH, the zwitterion of CMP may be considered as the initial form (or solution component). It should be noted that similar reasoning is valid for AMP and GMP. Figure 4 shows, as an example, the solution speciation for CMP depending on pH.

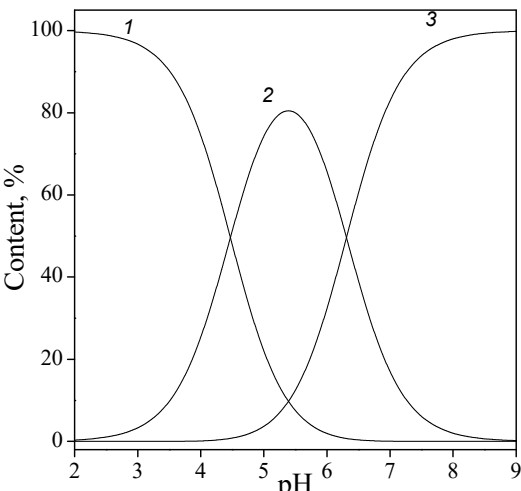

**Figure 4.** Solution speciation of cytidine-5′-monophosphate: zwitterion $H_2L^{\pm}$ (*1*), monoanion $HL^-$ (*2*), and dianion $L^{2-}$ (*3*).

Completely protonated forms of UMP and IMP were neutral, with anions $HL^-$ and $L^{2-}$ forming in the course of consecutive ionizations of phosphate residue.

The pyrimidine heterocycle of OMP contained an additional carboxyl group, which was ionized at pH > 2.4. Depending on the pH, the OMP molecule was present in the aqueous solution as completely protonated neutral particle $H_3L$ ($-PO_3H_2$, $-COOH$) or anions $H_2L^-$ ($-PO_3H^-$, $-COOH$), $HL^{2-}$ ($-PO_3H^-$, $-COO^-$), and $L^{3-}$ ($-PO_3{}^{2-}$, $-COO^-$).

At pH > 9, nitrogen atoms N3 in the pyrimidine ring of UMP and OMP were deprotonated with the formation of corresponding anions; however, the region of their existence was outside of the pH range in which the adsorption of these compounds has been studied.

Adsorption of nucleotides was studied depending on pH, ionic strength, and sorbent concentration. Figure 5 shows the adsorption of pyrimidine nucleotides as a function of titania concentration in the suspension. The degree of uptake is almost 100% at an oxide concentration of 5 g/L.

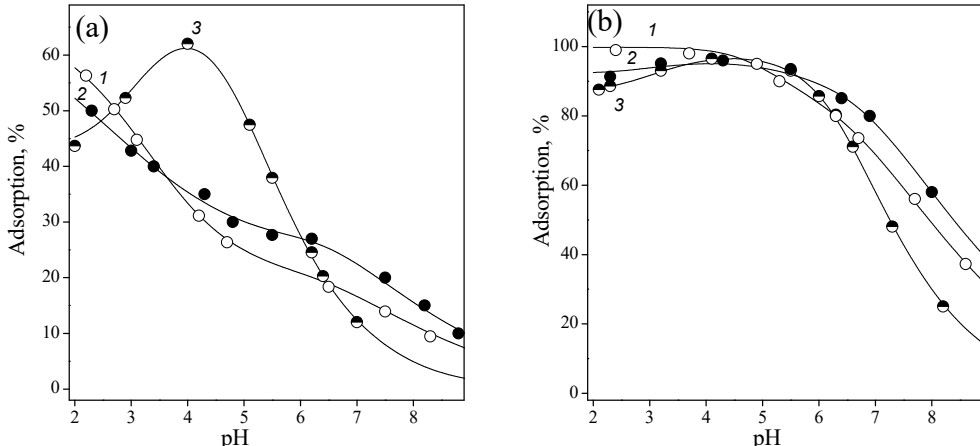

**Figure 5.** Adsorption of OMP (*1*), UMP (*2*), and CMP (*3*) from 0.01 M NaCl solution onto titanium dioxide surface: $C_{nucleotide}$ = 0.1 nmmol/L; $C_{TiO2}$ = 1 (**a**) and 5 (**b**) g/L.

As the concentration of $TiO_2$ decreased, the pH dependence of adsorption became pronounced and the difference of adsorption curves became more distinct. This is due to the fact that the initial particles that interacted with surface groups of titania had different charge signs. In addition to phosphate anions, which are common for all nucleotides, pyrimidine nucleotides contain the following charged groups: positively charged cytidine cycle, orotidine cycle bearing negative charge, and neutral uridine cycle.

In acidic medium, hydroxyl groups of titania were partially protonated ($\equiv TiOH_2^+$). It may be assumed that the negative charge of OMP favors adsorption, while the positive charge of CMP causes repulsion between similarly charged particles. Indeed, the adsorption of CMP increased with dissociation of the protonated nitrogen atom of the pyrimidine ring at pH > 4. As the pH further increased, the functional groups on the oxide surface were ionized ($\equiv TiO^-$). The decrease in the adsorption of all nucleotides at pH > 6 may be explained by the repulsion between similarly negatively charged anions and ionized surface groups.

Figure 6 shows adsorption of purine nucleotides from the aqueous solution as a function of pH.

The adsorption curves of the purine nucleotides under investigation differed only in the acidic region; after pH 5, they practically coincided. This was apparently due to the different basicity of the nitrogen atoms of the purine ring and the charge of the initial particle. Thus, for IMP, the initial component was a monoanion, and for AMP and GMP, they are zwitterionic forms that differed in the values of the constants for the deprotonation of nitrogenous bases. The presence of maxima on the adsorption curves of AMP and GMP suggests that particles with a positive charge on the nitrogen atom of the purine ring are not adsorbed on the surface of titanium oxide. An increase in adsorption was observed above a pH value approximately corresponding to the deprotonation

constant, as is usual at the adsorption of ionizable molecules [40]. The decrease in the adsorption of all nucleotides at pH > 6–7 can be explained by the repulsion of like-charged anions and ionized surface groups. Probably, the anionic forms of nucleotides interacted with the protonated hydroxyl groups of titanium dioxide.

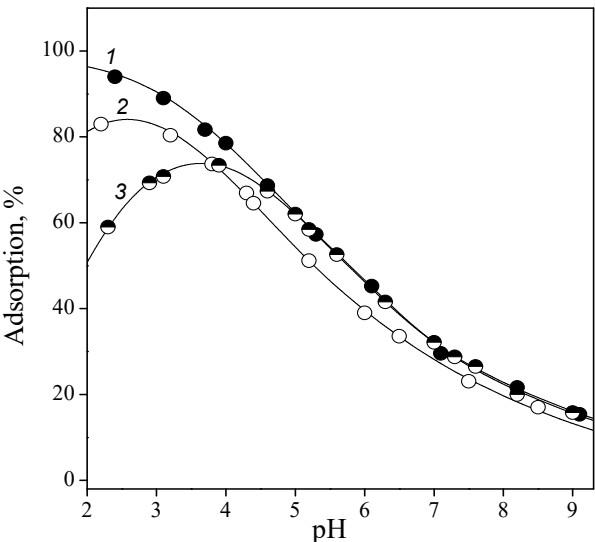

**Figure 6.** Adsorption of IMP (*1*), GMP (*2*), and AMP (*3*) from 0.01 M NaCl solution onto titanium oxide surface: $C_{nucleotide}$ = 0.1 nmmol/L, and $C_{TiO2}$ = 1 g/L.

For all nucleotides, adsorption curves could be modeled by two surface complexes with mono- and dianions. Only OMP charges of adsorbed species were different, namely, di- and trianions.

$$\equiv TiOH + H^+ + HL^- \leftrightarrow \equiv TiOH_2^+HL^-,$$

$$K_{1S}^{HL} = \frac{[TiOH_2^+HL^-]}{[TiOH][H^+][HL^-]} \exp(F(\Psi_0 - \Psi_1)/RT),$$

$$\equiv TiOH + HL^- \leftrightarrow \equiv TiOH_2^+L^{2-}$$

$$K_{2S}^{HL} = \frac{[TiOH_2^+L^{2-}]}{[TiOH][HL^-]} \exp(F(\Psi_0 - 2\Psi_1)/RT).$$

In the equations of equilibrium constants, the coefficients at the potentials ($\psi_0$ and $\psi_1$) correspond to the charges in the 0 and 1 planes. The charge of the protonated $\equiv TiOH_2^+$ group in the 0 plane is +1, and the charge in the 1 plane is determined by the charge of the nucleotide particle that forms the adsorption complex: −1 or −2. For OMP surface complexes, these values are −2 and −3, respectively.

The obtained equilibrium reaction constants can be recalculated into the formation constants of the complexes taking into account the equilibrium constants of the protonation reaction of the TiOH groups and the ionization/protonation reactions of the nucleotide:

$$\equiv TiOH_2^+ + HL^- \leftrightarrow \equiv TiOH_2^+HL^-,$$

$$\equiv TiOH_2^+ + L^{2-} \leftrightarrow \equiv TiOH_2^+L^{2-}.$$

The equilibrium reaction constants and formation constants of surface nucleotide complexes are presented in Table 4. The pyrimidine surface complexes were less stable than purine nucleotide complexes. Surface nucleotide complexes with a dianion were more stable than complexes with a monoanion. The same regularity was also observed for complex compounds of inorganic phosphate with some transition

metal ions: the stability constants of complexes Cu $(H_2PO_4)^+$ and Cu $(HPO_4)$ differed by 2 orders of magnitude [41], and for of Fe (III) ions, this difference was even larger [42].

**Table 4.** Equilibrium reaction and formation constants of nucleotide surface complexes (logK $\pm$ 0.05).

| Surface Reaction | Nucleotide | | | | | |
|---|---|---|---|---|---|---|
| | CMP | UMP | OMP | AMP | GMP | IMP |
| $\equiv TiOH + HL^- + H^+ \leftrightarrow \equiv TiOH_2^+ HL^-$ | 10.14 | 9.59 | | 10.60 | 10.31 | 10.11 |
| $\equiv TiOH_2^+ + HL^- \leftrightarrow \equiv TiOH_2^+ HL^-$ | 4.94 | 4.39 | | 5.40 | 5.11 | 4.91 |
| $\equiv TiOH + HL^- \leftrightarrow \equiv TiOH_2^+ L^{2-}$ | 4.29 | 4.06 | | 5.36 | 5.19 | 4.90 |
| $\equiv TiOH_2^+ + L^{2-} \leftrightarrow \equiv TiOH_2^+ L^{2-}$ | 5.24 | 4.90 | | 6.62 | 6.46 | 6.18 |
| $\equiv TiOH + HL^{2-} + H^+ \leftrightarrow \equiv TiOH_2^+ HL^{2-}$ | | | 9.75 | | | |
| $\equiv TiOH_2^+ + HL^{2-} \leftrightarrow \equiv TiOH_2^+ HL^{2-}$ | | | 4.55 | | | |
| $\equiv TiOH + HL^{2-} \leftrightarrow \equiv TiOH_2^+ L^{3-}$ | | | 6.32 | | | |
| $\equiv TiOH_2^+ + L^{3-} \leftrightarrow \equiv TiOH_2^+ L^{3-}$ | | | 6.22 | | | |

Apparently, outer-sphere complexes were formed, the components of which were connected by electrostatic bonds. This was confirmed by the dependence of the adsorption of all nucleotides on the ionic strength of the solution. For example, the adsorption of IMP as a function of ionic strength is shown in Figure 7. In the acidic region competition between phosphate anions and background electrolyte anions was manifested: the higher the concentration of the latter, the lower the adsorption of phosphate anions. With an increase in pH, adsorption from solutions with high ionic strength became larger compared with adsorption from less concentrated solutions. This was probably due to an increase in the concentration of protonated groups with an increase in the ionic strength of the solution.

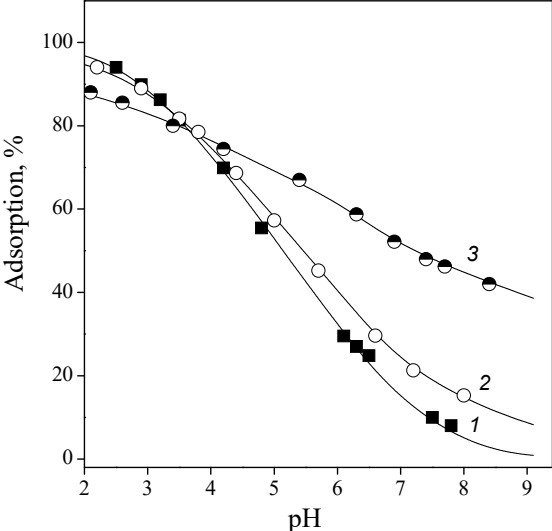

**Figure 7.** Adsorption of IMP on a titanium oxide surface as a function of ionic strength: 0.001 (*1*), 0.01 (*2*), and 0.1 (*3*) M NaCl. $C_{IMP}$ = 0.1 mmol/l, and $C_{TiO2}$ = 1 g/L.

Based on the calculated equilibrium reaction constants, the surface speciations of the complexes of the IMP as a function of pH and ionic strength were plotted (Figure 8). It should be noted that the curves reflecting the content of various complexes intersect at pH > 6, which corresponds to the ionization constant of a monoanion when it is converted into a dianion (i.e., ionization of the adsorbed single-charged anion on the surface occurs at approximately the same pH values as in the solution).

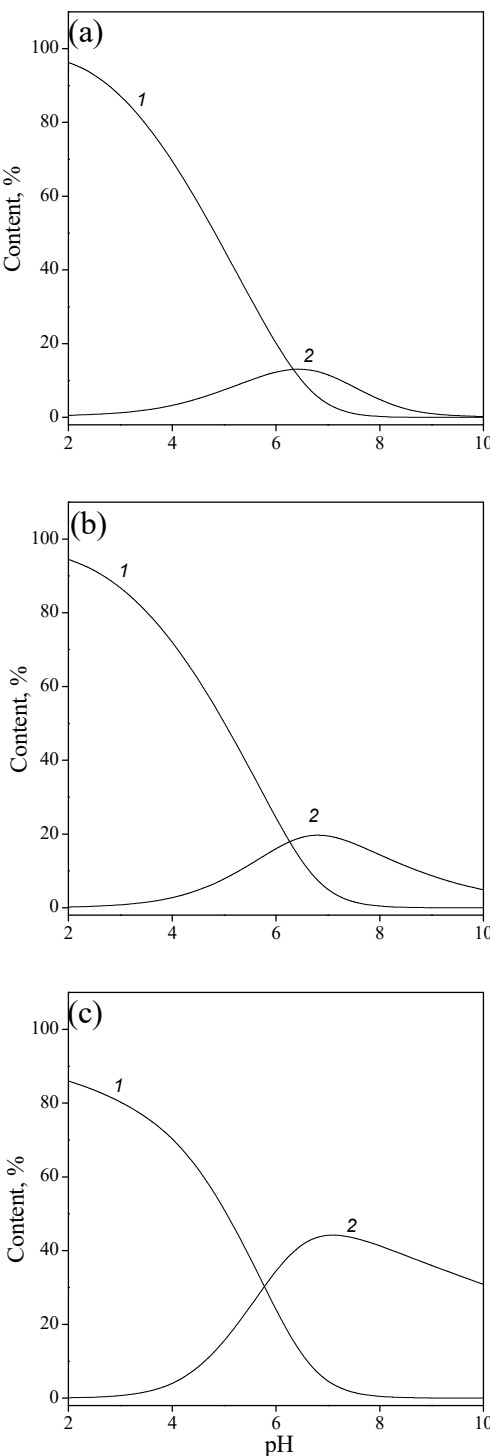

**Figure 8.** Surface speciation of adsorption complexes of IMP as a function of pH and ionic strength: 0.001 (**a**), 0.01 (**b**), and 0.1M NaCl (**c**); $\equiv TiOH_2^+ HL^-$ (*1*), and $\equiv TiOH_2^+ L^{2-}$ (*2*).

## 4. Conclusions

Interactions between aqueous nucleic acid components and mineral surfaces influence the bioavailability of these organic molecules in the environment, the viability of titanium implants in humans, and the role of mineral surfaces in the origin of life [43,44]. Considerable differences in the adsorption of nucleobases, nucleosides, and nucleotides with anatase surface were observed. The nucleotides formed more stable complexes onto the titanium oxide surface than nucleobases and

nucleosides. This can be explained by different types of interactions between sorbates and surface groups. Electrostatic interactions determining adsorption of nucleotides are stronger and conducted on a longer distance than hydrogen bonding and dispersion interactions which are characteristic of bases and nucleosides. The effect of heterocyclic bases on the stability of surface complexes is insignificant, which is apparently due to the participation of phosphate anions that are identical for all nucleotides in adsorption interactions. It can be assumed that nucleic acids, in which the active base groups are involved in the formation of hydrogen bonds and the phosphate groups are free, interact with the titanium oxide surface to form the same complexes as nucleotides.

**Author Contributions:** Conceptualization, N.V.; experiments, N.V. and O.M.; writing—original draft preparation, N.V.

**Funding:** This research received no external funding.

**Acknowledgments:** The authors are grateful to the M. Kersten (Johannes Gutenberg University of Mainz, Germany) for his help in purchasing materials used for experiments.

**Conflicts of Interest:** The authors declare no conflicts of interest.

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
