# Peer review of "Surface Complexation Modeling of Biomolecule Adsorptions onto Titania"

_colloids, doi:10.3390/colloids3010028_

Round 1
Reviewer 1 Report
In the scanned version of the manuscript you will find suggestions for changes and points numbered #i. These latter I discuss below.
#1. How did the authors make sure the water was CO2-free?
#2. This is not clear. What particles on what surface?
#3. They also correspond to the potentials at those planes. The electrostatic components are subject to a treatment different from the chemical components.
#4. There are papers that point to different interpretations. They would claim that the interfacial dielectric profile would tend to the value of the solid. In the case of anatase with its high static dielectric constant, this would change things completely. I think this statement could be made more objective.
#5. I think the term electrostatic bonds is not good. Why not speak about electrostatic attraction or call this ion-pairs?
#6. Not clear what is meant by row.
#7. Initial (and I think it pops up several times) is supposed to mean the surface without biomolecule adsorbed. I would rephrase those passages.
#8. Here particle refers to the solute, why not call it biomolecule?
#9. I think the maximum in the adsorption envelopes has been discussed in the very early papers on anion adsorption (paper in nature by Posner/Quirk if I remember correctly). Maybe a reference to this would be good, also supporting the idea that the maximum is at the pKa of the acid.
#10. Please be clear about stability constants and formation constants. Sometimes (here) it is better to use the latter term.
#11. This sentence left me a bit confused. I would suggest to rephrase it. Also state what high ionic strength is (for me it is above 2 M). Furthermore, it did not become clear to me which biomolecules were studied in different ionic strengths and what the outcome was. Maybe a table in the materials section stating what systems were studied would be helpful in this respect. The table could also include the pKa values of the biomolecules.
#12. Not clear what is stronger. The electrostatic contribution varies with pH and can become repulsive. H-bonds can be quite strong.
#13. The sentence made no sense to me.
In general I wondered where the accuracy of the log K values are coming from (note tha tin line 168, it should refer to the log of the equilibrium reaction constants).
Also at some point the authors stopped using equation numbers.
The authors might want to discuss the nature of the surface complexes (i.e. what spectroscopic evidence is available and what does it point to?) based e.g. on ATR-FTIR results.
Finally with respect to the modelling the authors use the somewhat old-fashioned two-pKa version of the surface complexation model. They might want to justfiy that choice.

Author Response
Thank you very much for attention to our paper. We tried to take into account all your comment and suggestions.

Reviewer 2 Report
I suggest that all characterization materials Table S1, Fig S1and S2 to be included in the body of the manuscript as they are important for the discussion of the results obtained.
Author Response
A lot of thanks for your attention to our paper.
Response: As concerned your suggestion: we have included all our tables and figures in the body of the manuscript.
Reviewer 3 Report
This manuscript studied and modeled the surface complexation of biomolecule adsorptions on to titania. Biomolecule adsorption on solid materials is important and has a broad impact in chemistry and biology research fields. The study suggests that there is considerable difference in the binding of nucleobases, nucleosides and nucleotides with anatase surface under a wide range pH. This is a good study, and I recommend to accept this manuscript if authors can answer the following questions.
1. pH value has an important impact on the chemical reactions studied in the manuscript. The pH value is quite different in solution and on solid surface based on the study of “capture CO2 from ambient air using nanoconfined ion hydration” Andewandte Chemie 128 (12), 4094-4097, and also the paper of “The effect of moisture on the hydrolysis of basic salts.” Chemistry–A European Journal, 22(51), pp.18326-18330. The two papers are helpful to readers understand the background of the pH value’s effect on the adsorption reaction on solid surface.
2. In figure 1, why the surface charge of No. 1 doesn’t decrease linearly with the increase of –log[H+]?
3. Eq. 5, should be separated to Eq. 5a and Eq. 5b, and also the equation of sigma, why there isn’t a number for the equation?
4. In figure 2, what is the meaning of question mark on Y axis?
5. In figure 3, why did authors use same marks for No.1 and No. 4?
6. Please give a number for each chemical equation
Author Response
Many thanks for your attention to our paper. We tried to answer on your questions.
